# Quantification of Polyphenols and Metals in Chinese Tea Infusions by Mass Spectrometry

**DOI:** 10.3390/foods9060835

**Published:** 2020-06-25

**Authors:** Gabriella Pinto, Anna Illiano, Andrea Carpentieri, Michele Spinelli, Chiara Melchiorre, Carolina Fontanarosa, Martino di Serio, Angela Amoresano

**Affiliations:** 1Department Chemical Sciences, University of Naples Federico II, Monte S. Angelo-Cinthia, 80126 Naples, Italy; gabriella.pinto@unina.it (G.P.); acarpent@unina.it (A.C.); mic.spinelli@studenti.unina.it (M.S.); chiara.melchiorre@unina.it (C.M.); carolina.fontanarosa@unina.it (C.F.); martino.diserio@unina.it (M.d.S.); angamor@unina.it (A.A.); 2Istituto Nazionale Biostrutture e Biosistemi-Consorzio Interuniversitario Viale delle Medaglie d’Oro, 305, 00136 Roma RM, Italy

**Keywords:** metabolites, polyphenols, heavy metals, mass spectrometry, MRM mode, Chinese tea infusion

## Abstract

Chemical compounds within tea *(Camellia sinensis)* are characterized by an extensive heterogeneity; some of them are crucial for their protective and defensive role in plants, and are closely connected to the benefits that the consumption of tea can provide. This paper is mainly focused on the characterization of polyphenols (secondary metabolites generally involved in defense against ultraviolet radiation and aggression by pathogens) and metals, extracted from nine Chinese tea samples, by integrating different mass spectrometry methodologies, LC-MS/MS in multiple reaction monitoring (MRM) and inductively coupled plasma mass spectrometry (ICP-MS). Our approach allowed to identify and compare forty polyphenols differently distributed in tea infusions at various fermentation levels. The exploration of polyphenols with nutraceutical potential in tea infusions can widely benefit especially tea-oriented populations. The worldwide consumption of tea requires at the same time a careful monitoring of metals released during the infusion of tea leaves. Metal analysis can provide the identification of many healthy minerals such as potassium, sodium, calcium, magnesium, differently affected by the fermentation of leaves. Our results allowed us: (i) to draw up a polyphenols profile of tea leaves subjected to different fermentation processes; (ii) to identify and quantify metals released from tea leaves during infusion. In this way, we obtained a molecular fingerprint useful for both nutraceutical applications and food control/typization, as well as for frauds detection and counterfeiting.

## 1. Introduction

During the growth, plants assimilates many nutrients from the soil through their roots and from the air through their leaves and other tissues. As with all the other living organisms, the heterogeneity of compounds required for the development of a plant is remarkable: macronutrients are constituted by lipids, glucides, and proteins, while micronutrients are mainly represented by minerals, vitamins, and polyphenols. *Camellia sinensis* is the plant whose leaves are used for the manufacturing of tea, one of the most consumed beverage all over the world, well known for its health-promoting intake, mainly correlated to the antioxidant activities of polyphenols [1,2]. Polyphenols (secondary metabolites generally involved in defense against ultraviolet radiation or aggression by pathogens [3]) are responsible for tea color, astringency, and flavor [4]. Moreover, polyphenols belong to a biologically active class of molecules characterized by multiple phenol functional groups (a hydroxyl group bonded to an aromatic ring) known for their antioxidative, antimutagenic, and anticarcinogenic effects [5]. The polyphenolic composition of tea leaves is strongly correlated to the type of tea e.g., green (unfermented), black (fully fermented), and blue (semi-fermented) [6]. Indeed, after harvesting, green tea leaves are rapidly steamed in order to inactivate the polyphenol oxidase, the enzyme capable of oxidizing tea polyphenols (such as catechins) to oligomeric and polymeric derivatives [7,8]. Instead, the fermentation process of black tea makes the polyphenols susceptible to enzymatic oxidation, allowing the conversion of catechins into polymeric compounds, such as theaflavins and thearubigins, which confer the characteristic aroma and color, typical of the black tea. Blue tea (oolong) undergoes a shorter fermentation process than black tea, therefore, catechins result to be less oxidized [9]. Green tea shows the highest quantity of tea catechins, chemically defined as flavan-3-ols, and of its gallate derivatives e.g., (−)-epigallocatechin gallate (EGCG), (−)-epicatechin gallate (ECG), (−)-epigallocatechin (EGC), and (−)-epicatechin (EC) [10,11]. Recently, the antioxidant power of catechins and their gallate derivatives were tested on HT29 cells, demonstrating that the esterification of a hydroxyl group at position 3 with gallic acid strongly decreases the values of standard reduction potentials displaying the higher antioxidant activity as the free radical scavenger [12].

The high potentiality of a tea beverage as a good antioxidant source (mediated mainly by polyphenols) jointly to the daily consumption worldwide suggests a more in-depth study on the risk of contamination by heavy metal absorbed by the plant via external environment (connected to farming habits and/or environmental pollution) [13,14]. These elements are important indicators for testing tea quality, as they can be transferred into tea infusions, then they can be assimilated by the human body through tea consumption, thus representing in potential risks to human health [14]. Some ions present in tea are healthy and their intake is recommended, but the co-existence of analytes (i.e., heavy metals) potentially risky for human consumption make their monitoring crucial. The development of a methodology capable of providing a wide molecular characterization of food is a crucial task for offering a potential tool of quality control and product typization [15,16].

Nowadays, the detection of polyphenols in food matrices is based on standardized methods based on colorimetric, e.g., Folin–Ciocalteu assay, and HPLC-UV analyses [17]. Recently, the quantification of eight catechins by ultra-HPLC coupled to triple quadrupole mass spectrometer (UHPLC-QqQ/MS) used in multiple reaction monitoring (MRM) has been performed in green tea [18] or in black tea infusions for an in vivo bioavailability study on tea-consuming guinea pigs [19]. The MRM approach is widely applied in proteomics [20] or metabolomics investigations [21].

This method was used for a targeted metabolomics approach by tandem mass spectrometry in MRM mode to selectively monitor several specific molecules belonging to the different classes of polyphenols. Target molecules were selected with their specific precursor and product ion transitions as defined by their fragmentation pattern described by their chemical structure. A single MRM method was devised to detect and quantify the target polyphenols within the samples leading to the unambiguous discrimination among the different tea infusions by a single analysis due to its high sensitivity, selectivity, and accuracy.

## 2. Materials and Methods

### 2.1. Materials

Solvents used for the sample preparation and LC/MS/MS analysis have a >99.9% of purity as reported by manufacturing companies. Acetic acid (CH_3_COOH) was purchased from Baker (Phillipsburg, NJ), acetonitrile (ACN), formic acid (HCOOH), and were from Honeywell (North Carolina, United States). Methanol (MeOH) were purchased from Sigma Aldrich (St. Louis, MO, USA). (-)-Epicatechin, gallic acid, vanillic acid, caffeic acid, and trans-p-coumaric acid used as standard to set up the quantification method were purchased from Sigma Aldrich (St. Louis, MO, USA). Nitric acid and hydrogen peroxide were purchased from Romil (Cambridge, UK).

### 2.2. Samples

Nine tea samples were selected for this study. All of them from China, seven were obtained from a Chinese local market. Particularly, six of them were black teas, two were green tea and one was oolong tea.

### 2.3. Metal Extraction of Tea Leaves

Tea leaves underwent a mineralization procedure as follow: 0.15 g were accurately weighed into glass tubes. Concentrated nitric acid (7 mL) and hydrogen peroxide (1 mL) were added to each sample. The reaction was conducted for 3.5 h at temperature of 90 °C. After the hydrolysis, the solution was filtered on 0.45 µm Millex filter (Millipore, Merck, MA, USA) and diluted to 25 mL with milli-Q water. Five mL of the solution were transferred into an ICP-MS vial for the analysis [22].

### 2.4. Metal Extraction of Tea Infusion

For the preparation of tea infusion, 400 milligrams of tea were weighed and added to 15 mL of gently boiling milli-Q water for four minutes, stirring during the infusion time. After the mixtures were filtered, each infusion (1 mL) was then mineralized as previously described and was analyzed by ICP-MS.

### 2.5. ICP-MS Analysis

Standard solutions have been prepared in 3% nitric acid at five different concentrations (0, 1, 10, 50, and 100 μg L^−1^). Metal concentrations have been measured with three replicates. Measurements were performed on an Agilent 7700 ICP-MS instrument (Agilent Technologies Santa Clara, CA, USA) equipped with a frequency-matching radio frequency (RF) generator and 3rd generation Octapole Reaction System (ORS3) operating with helium gas in ORF. The following parameters were used: RF power: 1550 W, plasma gas flow: 14 L min^−1^; carrier gas flow: 0.99 L min^−1^; He gas flow: 4.3 mL min^−1^. ^103^Rh isotope was used as an internal standard (final concentration: 50 μg L^−1^).

### 2.6. Sample Preparation for LC-MS/MS Analysis

#### 2.6.1. Sample Preparation

A set of nine tea samples differentiated in black and green tea were analyzed. Hot infusions were prepared by adding 10 mL of milli-Q water at about 85 °C to each sample and leaving to infuse for 4.5 min. Infusion was filtered on 0.45 µm Millex filter (Millipore, Merck) and the filtrate was diluted 1:1 with methanol acidified with 0.2% of acetic acid.

#### 2.6.2. Quantitative Analysis: External Standard Method

Quantitative analysis on tea samples was conducted with the external standard method by using vanillic acid, gallic acid, caffeic acid, p-coumaric acid, (-) epicatechin as standard. For each standard, a solution of 1 mg/mL in a 70% methanol, 30% water, and 0.1% acetic acid buffer was prepared in order to proceed with the serial dilutions ranging from 500 ppm to 0.25 ppb for building up calibration curves. Each point of the calibration curve was analyzed in triplicate by using the LC-MS/MS method in MRM ion mode. The upper limit of working range has been defined for each selected standard compound as the concentration where the instrumental response becomes non-linear.

Limit of detection (LOD) and limit of quantification (LOQ) were estimated by considering the standard deviation of y-intercept and the angular coefficient of the calibration curves calculated for each single standard. To evaluate repeatability and accuracy, 250 pg/µL standards’ mixture was analyzed 10 times (*n* = 10) on the same day and under the same conditions. Therefore, repeatability was calculated as relative standard deviation (%RSD). Accuracy was defined as: *%Accuracy = C_exp_/C_std_* × 100 where *C_exp_* was the quantification results and *C_std_* was the known concentration of the standard analyte.

The matrix effect was evaluated by spiking a mixture of standards into 100 mg of a pool of the tea infusions to reach a final concentration of each standard molecule of 500 pg/µL. This mixture was treated as reported for tea infusion samples to extract polyphenols content and then analyzed 10 times on the same day and under the same conditions.

### 2.7. LC-MS/MS Analysis

Polyphenols were separated by liquid chromatography using a LC Eksigent operating in microLiters per minute, with a column Halo C18 2.7 um 90A 1 × 50 mm, at a temperature of 38 °C. Eluent A was H_2_0 and 0.1% acetic acid, whereas eluent B was a solution on 50%ACN, 50% isopropanol and 0.1% acid acetic. Gradient conditions were as follows: time 0 min, 20% eluent B; time 4 min 90% eluent B; time 4.5 min 20% eluent B; time 5 min 20% eluent B. The injection volume was 5 μL and the flow rate was fixed to 40 μL/min. The mass spectrometer was a 4000 QTRAP^®^ of AB Sciex (Foster City, CA, USA), this system includes a hybrid triple-quadrupole LIT (Linear Ion Trap) designed for quantitative and qualitative analyzes. The study was carried out in MRM mode, typical “scan type”, with electrospray ionization source (ESI). Precursor and product ions were scanned in negative mode. The parameters for all the molecules monitored by optimized MRM method e.g., precursor (Q1) and product ions (Q2) as well as collision energy (CE) as well as declustering potential (DP) are listed in Appendix A. Source dependent parameters, ion spray voltage, source temperature, nebulizer gas, were set a 4.5 kV, 380 °C, and 25 psi, respectively.

### 2.8. PCA Analysis

Multivariate statistical analysis by using the principal component analysis (PCA) and heat maps were performed by XLStat 2016.5 version.

## 3. Results and Discussion

### 3.1. Optimization of LC-MS/MS Method for Polyphenols Quantification

Before proceeding with polyphenol extraction from tea leaves, standard solutions of vanillic acid, gallic acid, caffeic acid, p-coumaric acid, (-)-epicatechin were analyzed to set up the best instrumental parameters to be used for LC-MS/MS analysis in MRM mode (Appendix A). The instrumental response was recorded in triplicate for each analyte at the increasing concentration in the range of 1 to 500 ppb. Analytical parameters were determined and the LOD, LOQ, upper limit of working ranges, the calculated y-intercept, and the angular coefficient obtained for the calibration curves were reported in Table 1. Values of R^2^ tending to 1 indicated a strong correlation between concentration and instrumental response for each calibration curve.

Repeatability and accuracy for each standard are summarized in Table 1. These values were obtained by considering n = 10 replicate analysis on 250 pg/µL solution of standards’ mixture. Matrix effect was evaluated by spiking tea infusions with a known amount of a mixture of standards (250 pg/µL each). This spiked sample was treated as reported in the Material and Method section and then analyzed n = 10 times in the same condition of the standard samples and on the same day to evaluate the effect of interfering substances in standard quantification (Table 1).

### 3.2. Extraction of Polyphenols from Tea Infusions

Tea leaves were infused in semi-boiling water and an aliquot of each infusion was subjected to a liquid/liquid extraction by the addition of methanol acidified with acetic acid, as reported in the Material and Method section. Each sample was analyzed by LC-MS/MS analysis in MRM mode. The selection of the best transitions was designed by a combination of literature data and optimization of instrumental parameters [23,24]. Although several analytes eluted at similar retention times, it was possible to discriminate among the various molecules based on the peculiar transitions thanks to the great potentiality of the MRM ion mode. Indeed, it was possible to extrapolate from the TIC chromatogram any compound by selecting the opportune transitions and integrating the area under the relative peak. An example of TIC chromatogram for a green tea infusion sample was reported in Figure 1A where the epicatechin-3-O-gallate was among the most abundant detected ions. The extracted ion chromatogram (EIC) of one of the most abundant polyphenols i.e., the epicathecin-3-O gallate was shown (Figure 1B), whereas the qualifier (441 → 289 m/z) and quantifier (441 → 169 m/z) transitions were chosen for confirming identification and for estimating its concentration, respectively (Figure 1B).

An absolute quantification was carried out for all the standards mentioned above (Table 1). The values of the relative concentration were obtained by interpolating each calculated EIC peak area on the calibration curve [25]. Each concentration of polyphenols was expressed in µg/g, considering the volume of extraction, the tea leaves grams used for the infusions and the average recovery value (Table 1). For the other molecules for which the standards were not available, a relative quantification was performed by comparison of the areas underlying the EIC peaks for quantifier transitions. Although the use of each specific standard makes the analytical data more robust, a relative comparison of molecules along the different samples overpassed this bottleneck by using the same instrumental parameters.

Forty analytes were selected for targeted analyses, whereas a complete list of polyphenols detected in each tea infusions was collected in Appendix A.

Obtained data were visualized in a histogram representation to directly compare the trend of concentrations for each identified compound along the different composition of black, blue, and green tea infusions (Figure 2). The most abundant polyphenols observed in all samples were gallic acid and epicatechin 3-O-gallate, followed by two diastereomers i.e., epicatechin and catechin. The last two analytes showed the same instrumental response due to use of a non-chiral column that did not allow a chromatographic separation (Figure 2). Catechin/epicatechin were resulted to be more abundant in green tea than the other tea infusions, together with the other epicatechins derivatives, such as epicatechin-di-gallate, E-afzelechin (catechin with a missing hydroxyl group) gallate, and EGC-epicatechin dimer. The higher content of catechin and its gallate derivatives was also reported from others [26]. Indeed, the fermentation process occurring during the manufacturing of black and blue tea significantly affected the level of catechins in agreement with the exposure time of blue and black tea [11]. Therefore, the major nutraceutical compounds in green teas are catechins and the gallate derivatives for their well-known antioxidant activity [27].

Quinic acids derivatives, e.g., p-coumaroylquinic and chlorogenic acid, were more representative of green tea as previously reported by others [28], they drastically decreased in blue tea (Figure 2B). Instead, the most abundant phenolic acids in black tea were coumaric, protocatechuic, and caffeic acid, the most representative of coffee beans, whereas benzoic acid derivatives (for example of gallic acid) are more peculiar of teas [28]. Actually, the gallic acid was really abundant in black tea infusion jointly to the ethyl gallate (Figure 2A). These data are compatible with the fermentation process that remarkably increased its content [11]. Then, other polyphenols, e.g., diosmetin and theaflavin, were more abundant in black tea as a result of conversion of catechins into polymeric compounds during the fermentation. The exception was naringenin, doubly concentrated in green tea and absent in oolong tea (Figure 2C).

Among glucoside derivatives, rutin was detected at concentration of 60 µg/g in black tea, followed from 40 µg/g in oolong tea, and 30 µg/g in green tea infusion (Appendix A). Although the decrease of rutin from non-fermented to fully fermented has been previously connected with the fermentation process [29], others demonstrated that higher content of this metabolite together with catechin and quercetin was responsible for the characteristic flavor in black tea [30]. Even quercetin and its derivatives seemed not to be affected by the fermentation process [31]. Glucoside derivatives were in fact more abundant in black tea compared to other less fermented tea infusions (Appendix A). Quercetin is the major representative of flavonols, a subclass of flavonoids. It was only detected in black tea (Appendix A), together to its glycosidic derivatives, suggesting that this type of tea is a good source of differently absorbable quercetin. The interest for this flavonol is due to its wide range of bioactivity well documented by other authors [32,33,34]. Other polyphenols were only detected in black tea, such as vanillic acid and caffeine (Table 1 and Appendix A).

Differently from rutin and quercetin, kaempferol was more abundant in green tea as a rhamnoside form as shown also from others [35], while kaempferol-rutinoside was highly concentrated in black tea (Appendix A). A positive association between consumption of food containing kaempferol and reduced risk of occurrence of several disorders, such as cancer and cardiovascular diseases [36]. Moreover, kaempferol and its glucoside derivatives have a wide range of pharmacological and biological activities [36]. All glucoside derivatives were less abundant in oolong tea than in black and green teas (Appendix A), expect for myricetin-O-glucosides as reported previously from others [35].

Theobroma, the peculiar polyphenol of cocoa beans, was observed only in black and oolong tea leaves, probably responsible for the dark color and robust flavors of these tea leaves, together with theaflavins [9]. Actually, theaflavin is an orange-red pigment resulting from the oxidation and dimerization of catechins during the manufacture of black tea and oolong tea [9] and its higher content in these infusions has been suggested as responsible for increasing gut microbiota in short chain fatty acids (SCFA) generation, contrarily to role of unfermented green tea catechins [37].

While the chlorogenic acid content was higher in green tea, caffeine was detected only in black tea (Appendix A), differently from the data previously reported from others [11]. The variable content of this molecule did not seem to be correlated to the fermentation process, but rather to its instability and need of multiple extraction with aqueous methanol [11].

### 3.3. PCA Analysis

The results of PCA analysis of polyphenol levels in black infusion tea (samples numbered from 1–6), oolong tea (sample 7), and green tea (sample 8 and 9) were summarized in Appendix A.

The PCA representation of PC1 and PC2 factors allowed to clearly differentiate between the black tea infusions from those less fermented, except for the sample 4 that was grouped within the positive quadrant for both factors, similarly to blue and green teas (Figure 3). Anomalous clustering of sample 4 could be attributed either to a mixed composition of tea leaves or a peculiar treatment undergone by leaves.

The differentiation of polyphenol content reflected the molecular changes following the fermentation process as discussed above as evinced from the PCA values associated to gallic acid, (-)epicatechin/catechin, EC-3-O-gallate, and its derivates. The first of them showed a factor 1 at 14.620 and −6.875 while both positive values were associated to the factor 2 for the other ones. A similar observation can be attributed to the rutin as previously discussed (Figure 3).

### 3.4. Metal Quantification by ICP-MS

Further analysis was performed with the aim of the identification of metals, both in tea leaves and infusions by ICP-MS. This analytical technique has numerous advantageous features: large linear dynamic range, advantage of working with samples in solution, good precision, and low detection limits. The linear dynamic range for many elements is such that major, minor, and trace levels may all be determined at once. Furthermore, the multi-elemental analysis capability of ICP-MS allows the determination of many elements in a really short time.

The results were summarized in Appendix A, while a list of the average values (µg/g) of metals quantified in each tea type was reported in Table 2. The analyses were performed in triplicate and the average values of metals with concentration were expressed in μg/L and μg/g.

Tea is considered an important source on minerals such as manganese, an important element for human health, essential for development, metabolism, and the antioxidant system [38,39]. It seemed to increase as a result of the more intense fermentation process of leaves due to the highest content in black leaves (Figure 4). Manganese is an essential in detoxification of superoxide free radicals because it is contained in the enzyme superoxide dismutase (SOD), catalyzing the dismutation of superoxide radicals into either ordinary molecule oxygen (O_2_) and hydrogen peroxide (H_2_O_2_) [40]. Moreover, manganese can act as a neurotoxin in high quantity and its overexposure by an excessive ingestion or inhalation is associated neurological disorders [41]. Even the concentration of other metals was higher in black tea as a consequence of composition changes induced by fermentation with the exclusion of P, Mg, Ca, Na, Zn, Ba, Sr, Ni, Pb, Co, and Cd whose content was predominant in green tea (Figure 4 and Appendix A). Contrarily, these cited minerals were significantly affected by treatment of leaves. Interestingly, the blue teas contained the lowest concentration of all metals.

Moreover, other elements, particularly heavy metals as Pb, Cd, Cr, Co, Ni, Zn, Cu, which may be harmful to human health, are in lower concentration in both tea leaves and infusions (Appendix A). Their concentration ranged in the order of magnitude recorded in another study performed in the Chinese geographical area [35] and resulted lower than the values recently reported to be as not risky for human health [42].

## 4. Conclusions

A high number of target analytes were monitored and quantified by a single LC-MS/MS analysis in MRM ion mode, reaching a low limit of detection (up to 1.41 pg/μL). Among others, the content of catechins or gallic acid and their derivatives in infusion teas at different levesl of fermentation, e.g., green, blue, and black infusion teas, reflected the treatment undergone by leaves. Indeed, the fermentation process occurring during the manufacturing of black and blue tea significantly affected the level of catechins and their derivatives, whereas the gallic acid and ethyl-gallate were abundant in black infusion. Mineral quantification by combining the mineralization process to ICP-MS was carried out to follow their release from leaves into the tea infusions in order to confirm nutritional value of teas and to exclude the toxic level of heavy metals risky for human health in tea infusion samples. Essential micronutrients (e.g., potassium, sodium, calcium, magnesium, manganese, and phosphorus) were highly abundant in both tea leaves and infusions, especially in the black ones. On the other hands, the quantification of heavy metals potentially toxic for health resulted in concentrations lower than the values reported to be risky for humans.

The targeted MRM-MS and ICP analyses both gave back good results in terms of specificity and sensitivity for their applicability to food control toward a molecular fingerprinting useful for both nutraceutical applications and food typization, as well as in unmasking eventual counterfeiting and frauds.

## Figures and Tables

**Figure 1 foods-09-00835-f001:**
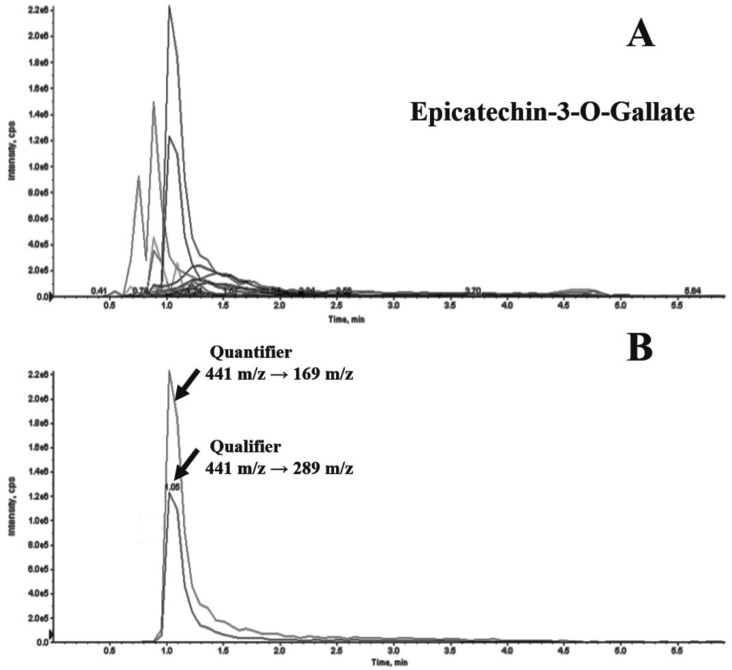
TIC chromatogram of a tea infusion sample (**A**) and MRM EIC of epicatechin 3-O-gallate (**B**). Quantifier 441 → 169 m/z and qualifier 441 → 289 m/z were reported.

**Figure 2 foods-09-00835-f002:**
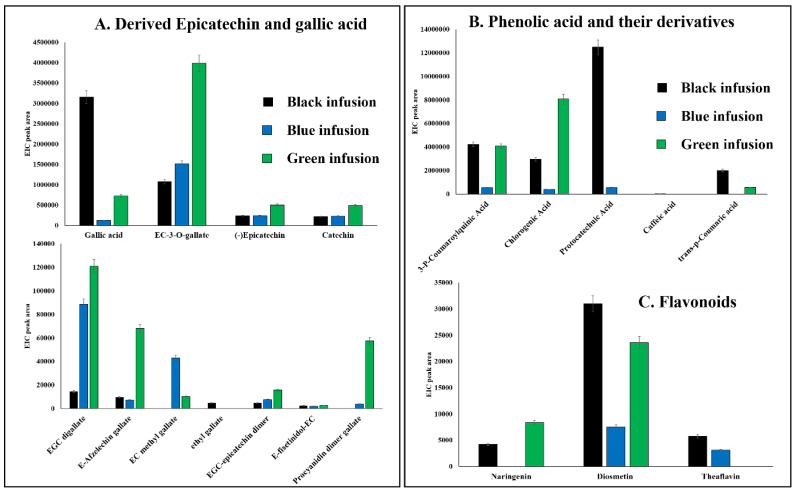
Histogram comparison of all polyphenols from tea infusion samples, i.e., black, oolong, and green teas grouped on the basis of its derivation form epicatechin/gallic acid (**A**), quinic acid (**B**), benzoic acid, and its derivatives (**C**) flavonoids. EC = epicathecin; EGC = epigallocatechin.

**Figure 3 foods-09-00835-f003:**
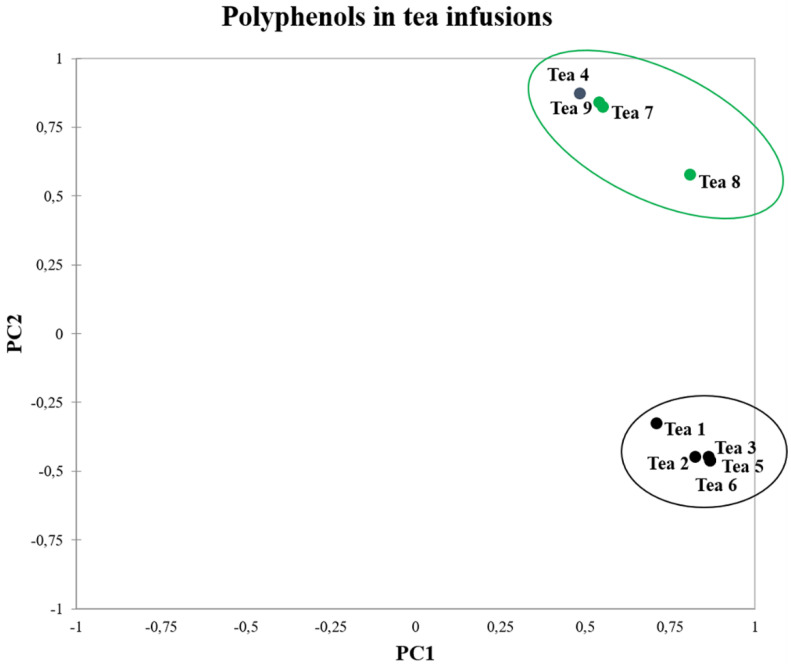
PCA analysis of tea infusions. 1–6 was related to the black teas while 7–9 was of oolong tea and green teas, respectively.

**Figure 4 foods-09-00835-f004:**
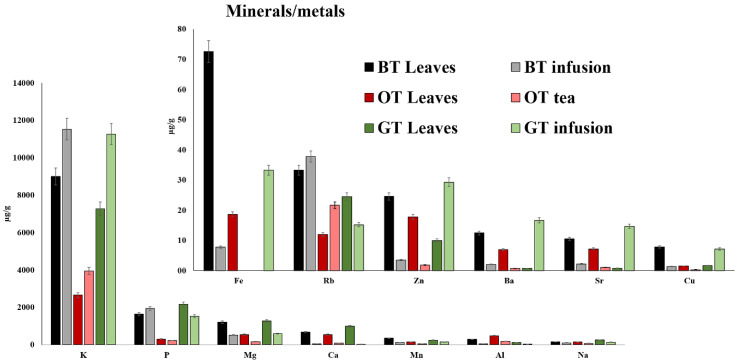
Histogram comparison of mineral and metal concentration in tea leaves and infusions of black tea (BT), blue tea (BT), green tea (GT).

**Table 1 foods-09-00835-t001:** Limit of detection (LOD), limit of quantification (LOQ), repeatability (%RSD), accuracy (%), matrix effect (%Recovery) calculated for six standard molecules, upper limit of working range (pg/µL), y-intercept, angular coefficient for the calculated calibration curves obtained for each standard compound are reported.

STANDARD	Quantifier m/z	m	q	R^2^	LOD (pg/µL)	LOQ (pg/µL)	Upper Working Range [pg/µL]	%RSD	%Accuracy	%Recovery	Black Infusion (µg/g)	Blue Infusion (µg/g)	Green Infusion (µg/g)
Gallic Acid	124.90	2996.5	76531	0.9935	15.16	45.95	500	17.1	85.20–126.14	87.2	582.43	22.95	159.16
Vanillic Acid	107.80	2314.6	15272	0.9983	7.69	23.31	500	3.69	91.70–98.8	92.3	4.31	-	-
Trans-p-Coumaric Acid	119.00	32776	291492	0.9986	1.41	4.3	500	3.19	97.13–103.72	95.5	10.15	-	3.48
Caffeic Acid	134.90	15148	137495	0.9993	4.14	12.55	500	3.18	87.73–94.60	89.4	1.01	-	1.01
(-)-Epicatechin	244.80	1200.2	1080.6	0.9982	3.25	9.84	500	14.6	75.96–101.62	85.2	32.62	38.52	83.95

Recovery values, calculated as described in the Material and Methods section, were reported in Table 1.

**Table 2 foods-09-00835-t002:** Metals quantification in leaves and infusion of three tea differently fermented (black tea = BT; blue tea = BL; green tea = GT). The average values were calculated for each tea type.

	Average Black Tea		Average Oolong Tea		Average Green Tea	
	BT Leaves	BT Infusion	BL Leaves	BL Tea	GT Leaves	GT Infusion
Metals	Conc. (µg/g)	Conc. (µg/g)	Conc. (µg/g)	Conc. (µg/g)	Conc. (µg/g)	Conc. (µg/g)
K	8996.102	11535.429	2655.626	3946.741	7264.210	11260.935
P	1644.388	1944.327	304.034	220.817	2165.274	1533.009
Mg	1216.800	515.621	544.392	170.475	1275.141	598.120
Ca	670.392	55.530	542.526	77.562	983.943	26.662
Mn	357.941	124.278	140.959	46.335	244.590	141.309
Al	287.661	49.306	483.618	182.910	115.614	34.697
Na	159.809	105.778	139.745	72.741	252.535	136.846
Fe	72.556	7.749	18.575	<0.001	33.254	<0.001
Rb	33.233	37.755	11.932	21.611	15.155	24.513
Zn	24.552	3.489	17.818	1.856	29.256	9.958
Ba	12.433	2.027	6.912	0.743	16.633	0.729
Sr	10.471	2.176	7.181	1.002	14.670	0.788
Cu	7.859	1.349	1.500	0.307	7.153	1.673
Ni	4.381	1.577	0.881	0.467	8.894	11.800
Cr	3.978	<0.001	0.487	<0.001	0.833	<0.001
Ti	2.125	0.201	0.503	0.033	1.518	0.097
Pb	0.281	0.069	0.138	0.009	0.679	0.027
Cs	0.219	0.203	0.035	0.112	0.071	0.088
Mo	0.165	<0.001	<0.001	<0.001	<0.001	<0.001
V	0.130	0.008	0.029	0.001	0.071	0.002
Co	0.121	0.041	0.017	0.026	0.213	0.373
LI	0.116	0.051	0.070	0.071	0.139	0.034
As	0.096	0.035	0.015	0.005	0.046	0.025
Sn	0.053	0.005	0.053	0.004	0.091	0.006
Sb	0.045	0.002	0.012	<0.001	0.035	0.008
Se	0.031	0.007	0.025	0.010	0.025	0.008
Cd	0.023	0.003	0.015	0.001	0.035	0.002
Be	0.004	0.002	<0.001	0.009	0.003	0.001
Te	0.001	0.000	0.003	0.001	0.001	<0.001

Our results clearly indicate that potassium is the most representative element in both tea leaves and infusions and its concentration increased probably due to infusion process occurring at the time-temperature couples reported in the Material and Method section (Table 2, Figure 4). Other ions e.g., phosphorus, magnesium, calcium, and manganese were similarly abundant with decreasing concentration along the infusion process excluded for phosphorus in black tea (Figure 4). Actually, the chemical elements, namely potassium, sodium, calcium, magnesium, manganese, and phosphorus, were the most abundant ones and their presence in food is considered as healthy and safe for the organism due to their involvement in the main cellular processes.

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
