# Peer review of "Quantification of Polyphenols and Metals in Chinese Tea Infusions by Mass Spectrometry"

_foods, 2020, doi:10.3390/foods9060835_

Round 1
Reviewer 1 Report
I have found this manuscript no interesting from the analytical view. Due to the fact that the polyphenols have been already studied very times by this method, and therefore, it is not novel. Besides, the extraction of these compounds from the tea has not been optimized; the conditions have been chosen of a previous work used for chocolates samples and the matrix is completely different. Therefore, the manuscript is not novel either in this point. I think it could be improved sharply recalculating the recovery for this kind of sample.
However, this manuscript seems interesting by the results obtained. They are very good explained and showed.
There are some mistakes that I would like to change:
1) “on” should be deleted from paragraph abstract line 17
2) abstract should be rewritten, it is not well expressed
3) Introduction the 70-78 line should be deleted due to the fact that in this paragragh your results should not be shown and only the intention of this study should be explain.
4) 2.3 paragraph the tittle should be changed to metal extraction of tea leaves
5) 99-100 line should delete
6) 2.4 paragraph the tittle should be changed to metal extraction of tea infusion
7) 2.6 paragraph, MRM analysis should be changed to sample preparation for LQ-MS/MS analysis
8)124 line LQ-MRM/MS analysis should be changed by LQ-MS/MS
9) Equations of LOD and LOQ should be deleted. They are not relevant information
10) Equations of RSD and accuracy should be deleted. They are not relevant information
11) The recovery was calculated for chocolate, I suppose that it was a previous work, but I think that you should have done this for your tee samples due to the fact that they are two matrixes completely different.
12) The recovery formula should be deleted.
13) 165 line, title MRM method optimization for polyphenols quantification should be change to optimization of LQ-MS/MS for polyphenols quantification
14) The figure 2 and 3 must be improved. I cannot see them clearly.
15) The commas should be substituted by points in both figures
16) 298 line, heavy must be deleted.
17) The range of limit of detection should be included in the conclusions
Author Response
Naples, 3rd June 2020
Dear Editor,
Please find enclosed the revised version of paper entitled “Quantification of polyphenols and heavy metals in Chinese tea infusions by mass spectrometry” (Manuscript ID: foods-817165) we amended according to referee’s suggestions.
We also enclose our replies to the Reviewers comments. The requested changes are highlighted in red along the tracked version of the revised manuscript.
We express our gratitude to you and to both Reviewers for your combined efforts aimed to improve our work.
We hope that the current version of our paper is now suitable for publication in Foods.
With Best Regards,
Anna Illiano, PhD
Depertment of Chemical Science
Complesso Universitario di Monte Sant’Angelo
Via Cynthia 4, I-80126 Napoli
Tel. + 39 081 674050 - Fax + 39 081 674313
e-mail: anna.illiano@unina.it
Detailed point-by-point reply to the Reviewers’ concerns
Reviewer 1
I have found this manuscript no interesting from the analytical view. Due to the fact that the polyphenols have been already studied very times by this method, and therefore, it is not novel. Besides, the extraction of these compounds from the tea has not been optimized; the conditions have been chosen of a previous work used for chocolates samples and the matrix is completely different. Therefore, the manuscript is not novel either in this point. I think it could be improved sharply recalculating the recovery for this kind of sample.
However, this manuscript seems interesting by the results obtained. They are very good explained and showed.
Author: Actually, since several years the group is focusing on the polyphenol quantification from several food matrices. A wide panel of molecules has been implemented for their quantification in two main matrix: chocolate and tea. Although two separate investigations proceeded in parallel, all the analytical parameters were calculated for each matrix separately. What was reported in the main text about chocolate was clearly a typo error we have just changed.
There are some mistakes that I would like to change:
- “on” should be deleted from paragraph abstract line 17
Author: The correction has been done.
- abstract should be rewritten, it is not well expressed
Author: The abstract has been rewritten
- Introduction the 70-78 line should be deleted due to the fact that in this paragragh your results should not be shown and only the intention of this study should be explain.
Author: 70-78 lines have been deleted.
- 3 paragraph the tittle should be changed to metal extraction of tea leaves
Author: The paraghaph title has been changed.
- 99-100 line should delete
Author: line 99-100 has been deleted.
- 4 paragraph the tittle should be changed to metal extraction of tea infusion
Author: The paraghaph title has been changed.
- 6 paragraph, MRM analysis should be changed to sample preparation for LQ-MS/MS analysis
Author: The paraghaph title has been changed.
- 124 line LQ-MRM/MS analysis should be changed by LQ-MS/MS
Author: the text has been modified.
- Equations of LOD and LOQ should be deleted. They are not relevant information
Author: The text was modified accordingly to reviewer suggestions.
10) Equations of RSD and accuracy should be deleted. They are not relevant information
Author: The text was modified accordingly to reviewer suggestions.
11) The recovery was calculated for chocolate, I suppose that it was a previous work, but I think that you should have done this for your tee samples due to the fact that they are two matrixes completely different.
Author: We apology for the misunderstanding due to typos error. The reported recovery values were calculated for infusion samples. We corrected the typos errors.
12) The recovery formula should be deleted.
Author: The text was modified accordingly to reviewer suggestions.
13) 165 line, title MRM method optimization for polyphenols quantification should be change to optimization of LQ-MS/MS for polyphenols quantification
Author: the title has been changed.
14) The figure 2 and 3 must be improved. I cannot see them clearly.
Author: Figure 2 and 3 have been modified. Some histogram representations have been moved to Supplementary section as SF 1 and 2.
15) The commas should be substituted by points in both figures
Author: The suggested changes have been done.
16) 298 line, heavy must be deleted.
Author: The text has been modified.
17) The range of limit of detection should be included in the conclusions
Author: LOD values were included to conclusions.

Reviewer 2 Report
This study was well carried out with the exception of some minor issues. I felt that the conclusions of this work are not of any interest particular to the community, given that a lot of work, and better carried out studies, exsist on teas; however, it is worth having this work published for the sake of having the results available for the record.
My major concern with this work is the lack of samples and improper statistical back sampling of teas within the study. The authors seems to have selected different numbers of teas by variety which makes it difficult to judge the significance of any of the conclusions given that a much more thorough statistical treatment could have been performed if sample size considerations were made.
The authors also need to clean up the English of this manuscript.
Author Response
Naples, 3rd June 2020
Dear Editor,
Please find enclosed the revised version of paper entitled “Quantification of polyphenols and heavy metals in Chinese tea infusions by mass spectrometry” (Manuscript ID: foods-817165) we amended according to referee’s suggestions.
We also enclose our replies to the Reviewers comments. The requested changes are highlighted in red along the tracked version of the revised manuscript.
We express our gratitude to you and to both Reviewers for your combined efforts aimed to improve our work.
We hope that the current version of our paper is now suitable for publication in Foods.
With Best Regards,
Anna Illiano, PhD
Depertment of Chemical Science
Complesso Universitario di Monte Sant’Angelo
Via Cynthia 4, I-80126 Napoli
Tel. + 39 081 674050 - Fax + 39 081 674313
e-mail: anna.illiano@unina.it
Detailed point-by-point reply to the Reviewers’ concerns
Reviewer 2
This study was well carried out with the exception of some minor issues. I felt that the conclusions of this work are not of any interest particular to the community, given that a lot of work, and better carried out studies, exsist on teas; however, it is worth having this work published for the sake of having the results available for the record.
My major concern with this work is the lack of samples and improper statistical back sampling of teas within the study. The authors seems to have selected different numbers of teas by variety which makes it difficult to judge the significance of any of the conclusions given that a much more thorough statistical treatment could have been performed if sample size considerations were made.
The authors also need to clean up the English of this manuscript.
Author: Conclusion paragraph has been rewritten to underline the relevance of our study. This paper represent a this preliminary study and the numbers of teas has been limited to the few ones found in a Chinese market. A further investigation can be extended to a wider sampling. English language has been revised along the main text.

Reviewer 3 Report
Pinto et al. analyzed Chinese tea infusions by mass spectrometry-based quantification methods and found characteristic profiles of polyphenols and metals as revealed by the comparison of constituent chemicals and by PCA analysis of tea samples. There are so many points that are not clear and/or needed to be corrected, and therefore, this reviewer cannot recommend the publication. Specific points are as follows.
- Please correct or rewrite the following words/expressions: heavy metals (in Title); to prevent counterfeiting and frauds (line 15); Mass Spectrometry/Multiple Reaction Monitoring (use lower case; line 18); LC-MS/MS/ICP-MS/MRM (spell out); healthy species (line 21); lower that (line 23); glucides/micronutrient (line 31); UHPLC-QqQ/MS (spell out; line 66); After the mixtures were filtered with filter paper. (lines 102-103); Measurement have been (lines 107-108); 103Rh (What is this?; line 111); Methanol/Water/Acetic Acid (use lower case; line 122); ESI (line 156); Precursor ion was scanned in negative mode. (lines156-157); a green infusion (tea?; line 194); Supporting Information Table 2. (Table S2; line 213); two di diastereomers (line 217); epicatechin di gallate (line 221); glyosidic (line 251); The PCA analysis obtained by (line 283); Mn-SOD (line 321); O2 (line 322); Physiological and Molecular Plant Pathology (journal names in upper case, see others too; lines 371-372); TANAKA, T.; KOUNO, I. (line 381). Please check more.
- Introduction.
2-1. Please explain what MRM analysis is.
2-2. The last sentence (lines 75-78) is not Introduction.
- Materials and Methods. Please indicate company names (and their locations) for instruments and software.
- Results and Discussion.
4-1. Although quantitative analysis was done only for several compounds, many compounds were quantified. Why?
4-2. “3.2. Extraction of polyphenols from tea infusions”. Lengthy descriptions about the contents and the effects of polyphenols are nothing new, and should be deleted.
4-3. What is the point in the PCA analysis in Figure 3? The purpose of this analysis is not clear.
4-4. This reviewer cannot understand why polyphenols and metals are discussed together here, as they are not really related, representing different methods and different physiological effects.
- Conclusions. The description in this section is totally unfounded. What are “great results” (line 353) shown here?
Author Response
Naples, 3rd June 2020
Dear Editor,
Please find enclosed the revised version of paper entitled “Quantification of polyphenols and heavy metals in Chinese tea infusions by mass spectrometry” (Manuscript ID: foods-817165) we amended according to referee’s suggestions.
We also enclose our replies to the Reviewers comments. The requested changes are highlighted in red along the tracked version of the revised manuscript.
We express our gratitude to you and to both Reviewers for your combined efforts aimed to improve our work.
We hope that the current version of our paper is now suitable for publication in Foods.
With Best Regards,
Anna Illiano, PhD
Depertment of Chemical Science
Complesso Universitario di Monte Sant’Angelo
Via Cynthia 4, I-80126 Napoli
Tel. + 39 081 674050 - Fax + 39 081 674313
e-mail: anna.illiano@unina.it
Detailed point-by-point reply to the Reviewers’ concerns
Reviewer 3
Pinto et al. analyzed Chinese tea infusions by mass spectrometry-based quantification methods and found characteristic profiles of polyphenols and metals as revealed by the comparison of constituent chemicals and by PCA analysis of tea samples. There are so many points that are not clear and/or needed to be corrected, and therefore, this reviewer cannot recommend the publication. Specific points are as follows.
- Please correct or rewrite the following words/expressions: heavy metals (in Title); to prevent counterfeiting and frauds (line 15); Mass Spectrometry/Multiple Reaction Monitoring (use lower case; line 18); LC-MS/MS/ICP-MS/MRM (spell out); healthy species (line 21); lower that (line 23); glucides/micronutrient (line 31); UHPLC-QqQ/MS (spell out; line 66); After the mixtures were filtered with filter paper. (lines 102-103); Measurement have been (lines 107-108); 103Rh (What is this?; line 111); Methanol/Water/Acetic Acid (use lower case; line 122); ESI (line 156); Precursor ion was scanned in negative mode. (lines156-157); a green infusion (tea?; line 194); Supporting Information Table 2. (Table S2; line 213); two di diastereomers (line 217); epicatechin di gallate (line 221); glyosidic (line 251); The PCA analysis obtained by (line 283); Mn-SOD (line 321); O2 (line 322); Physiological and Molecular Plant Pathology (journal names in upper case, see others too; lines 371-372); TANAKA, T.; KOUNO, I. (line 381). Please check more.
Author: the suggested correctionss have been done.
- Introduction.
2-1. Please explain what MRM analysis is.
Author: Some information related to MRM have been added to the main text.
2-2. The last sentence (lines 75-78) is not Introduction.
Author: The last sentence has been deleted.
- Materials and Methods. Please indicate company names (and their locations) for instruments and software.
Author: Company names and their locations were added to the main text.
- Results and Discussion.
4-1. Although quantitative analysis was done only for several compounds, many compounds were quantified. Why?
Author: The experience gained from a previous study on chocolate samples (under revision) suggested us to expand the investigation to the complete panel to extract a clearer profile of polyphenols in tea infusion. However, the availability of standards for the entire set of target molecules in a very difficult task. So we exploited the great potential of MRM method based on the simultaneous detection of a wide panel of molecules in a single run in a very short time. An robust absolute quantification was carried out for the molecules having the corresponding standards by interpolating each calculated EIC peak area on the calibration curve. For all the other molecules for which the standards were not available a relative quantification was performed by comparison the areas underlying the EIC peaks in order to obtain a general view of polyphenols in the different samples. Actually, the results on the quantification of main catechin and gallate derivatives were implemented with profiles of glucoside derivates that have more significant implication on their bioavailability.
4-2. “3.2. Extraction of polyphenols from tea infusions”. Lengthy descriptions about the contents and the effects of polyphenols are nothing new, and should be deleted.
Author: the main text has been modified and streamlined in accordance with the referee’s suggestions.
4-3. What is the point in the PCA analysis in Figure 3? The purpose of this analysis is not clear.
Author: although the limited number of available samples, PCA analysis results allowed to cluster the different types of tea at different fermentation processes based on the polyphenol content.
4-4. This reviewer cannot understand why polyphenols and metals are discussed together here, as they are not really related, representing different methods and different physiological effects.
Author: The exploration of a molecule class like polyphenols with nutraceutic potential in tea infusions can widely benefit especially the tea oriented population. In any case, the wide consumption of tea everywhere can not prescind from the investigation on monitoring of minerals to guarantee their content below that recommended. Based on that, several micronutrients have been detected and quantified by ICP-MS.
- Conclusions. The description in this section is totally unfounded. What are “great results” (line 353) shown here?
Author: Conclusion section has been changed.

Round 2
Reviewer 1 Report
In general, the manuscript has been improved.
Author Response
Reviewer 1:
Open Review
(x) I would not like to sign my review report
( ) I would like to sign my review report
English language and style
(x) Extensive editing of English language and style required
( ) Moderate English changes required
( ) English language and style are fine/minor spell check required
( ) I don't feel qualified to judge about the English language and style
Yes |
Can be improved |
Must be improved |
Not applicable |
|
Does the introduction provide sufficient background and include all relevant references? |
(x) |
( ) |
( ) |
( ) |
Is the research design appropriate? |
(x) |
( ) |
( ) |
( ) |
Are the methods adequately described? |
( ) |
(x) |
( ) |
( ) |
Are the results clearly presented? |
(x) |
( ) |
( ) |
( ) |
Are the conclusions supported by the results? |
(x) |
( ) |
( ) |
( ) |
Comments and Suggestions for Authors
In general, the manuscript has been improved.
Authors: the authors express their gratitude to the Reviewer 1 in considering the last version of the main text improved.

Reviewer 3 Report
This reviewer cannot find the revision of the manuscript enough to be a publication level. There are still so many errors in English in the text and references, and no satisfactory answers or responses to the previous comments. Especially, from the response to the comment #4-1, the authors are trying to skip “a very difficult task” to estimate the quantity of constituents by referring to “a general view of polyphenols”, but saying the method to give “good results in terms of specificity and sensitivity” (Conclusions). No data have been provided to examine this method is good or bad. For example, the analysis of catechin, by comparing it with a standard, whose characteristics are not clear in this manuscript, is explained in Fig. 1, although there is no comparison of the data with those by other quantification methods, which is needed to examine specificity and sensitivity. As to the PCA analysis in Fig. 3, what is the interpretation of the result? Is the conclusion that Tea 4 is not a black tea? If this analysis is good for distinguishing black tea from other teas, the best way is just to look at the teas and check how they smell. No need of PCA analysis. Conclusively, the authors took more space to discuss their bioavailability because the quantification of polyphenols is not well done.
Author Response
Reviewer 3:
Open Review
(x) I would not like to sign my review report
( ) I would like to sign my review report
English language and style
(x) Extensive editing of English language and style required
( ) Moderate English changes required
( ) English language and style are fine/minor spell check required
( ) I don't feel qualified to judge about the English language and style
Yes |
Can be improved |
Must be improved |
Not applicable |
|
Does the introduction provide sufficient background and include all relevant references? |
( ) |
( ) |
(x) |
( ) |
Is the research design appropriate? |
( ) |
( ) |
(x) |
( ) |
Are the methods adequately described? |
( ) |
( ) |
(x) |
( ) |
Are the results clearly presented? |
( ) |
( ) |
(x) |
( ) |
Are the conclusions supported by the results? |
( ) |
( ) |
(x) |
( ) |
Comments and Suggestions for Authors
This reviewer cannot find the revision of the manuscript enough to be a publication level. There are still so many errors in English in the text and references, and no satisfactory answers or responses to the previous comments.
Authors: English language was modified along the main text.
Especially, from the response to the comment #4-1, the authors are trying to skip “a very difficult task” to estimate the quantity of constituents by referring to “a general view of polyphenols”, but saying the method to give “good results in terms of specificity and sensitivity” (Conclusions).
Authors performed an absolute quantification for standard molecules providing all the analytical parameters (Table 1). Regarding the other polyphenols monitored in the MRM method, only a relative comparison was performed. Peak areas for all the species were compared to get a trend of metabolite expression in the different samples. Anyway promising results by MRM methodology were obtained taking advantage from the specificity of molecule identification based on fragmentation reactions in respect of colorimetric/spectrophotometric assays (es Folin-Ciocâlteu or HPLC)
No data have been provided to examine this method is good or bad. For example, the analysis of catechin, by comparing it with a standard, whose characteristics are not clear in this manuscript, is explained in Fig. 1, although there is no comparison of the data with those by other quantification methods, which is needed to examine specificity and sensitivity.
Author: Analytical parameters were obtained to test the MRM method performance and were included in the main text (Table 1). The working range for catechin was 4-500 pg/μL while those published for other approaches were 3-300 mg/L or 0,8-50 mg/L for HPLC and 0,3-5 mg/L UV-Vis spectrophotometric method.
(Determination of total catechins in tea extracts by HPLC and spectrophotometry Qiang He, Kai Yao, Dongying Jia Haojun Fan, Xuepin Liao and Bi Shi ; Sensitive Determination of Catechins in Tea by HPLC Thermo Application Note 275).
As to the PCA analysis in Fig. 3, what is the interpretation of the result? Is the conclusion that Tea 4 is not a black tea? If this analysis is good for distinguishing black tea from other teas, the best way is just to look at the teas and check how they smell. No need of PCA analysis.
Author:The PCA representation offers the opportunity to better visualize the data based on the expression level of the entire dataset of molecules. Actually, a good clustering of the different teas at different level of fermentation was obtained supporting the similarity in the molecular composition with the exception of the tea 4 undergone to the peculiar treatment (nor reported in the label), resulted to be more close to the green tea although it was a black tea. A sentence has been included in the main text.
Conclusively, the authors took more space to discuss their bioavailability because the quantification of polyphenols is not well done.
Author: the most of discussion about bioavailability of some molecules has been already removed during the first revision
